# Milk Pathogens in Correlation with Inflammatory, Oxidative and Nitrosative Stress Markers in Goat Subclinical Mastitis

**DOI:** 10.3390/ani12233245

**Published:** 2022-11-23

**Authors:** Cristiana Ștefania Novac, George Cosmin Nadăș, Ioana Adriana Matei, Cosmina Maria Bouari, Zsuzsa Kalmár, Smaranda Crăciun, Nicodim Iosif Fiț, Sorin Daniel Dan, Sanda Andrei

**Affiliations:** 1Department of Microbiology, Immunology and Epidemiology, Faculty of Veterinary Medicine, University of Agricultural Sciences and Veterinary Medicine of Cluj-Napoca, 400372 Cluj-Napoca, Romania; 2Department of Infectious Diseases, “Iuliu Hatieganu” University of Medicine and Pharmacy Cluj-Napoca, 400012 Cluj-Napoca, Romania; 3ELKH-ÁTE Climate Change: New Blood-Sucking Parasites and Vector-Borne Pathogens Research Group, 1078 Budapest, Hungary; 4Department of Animal Production and Food Safety, Faculty of Veterinary Medicine, University of Agricultural Sciences and Veterinary Medicine of Cluj-Napoca, 400372 Cluj-Napoca, Romania; 5Department of Biochemistry, Faculty of Veterinary Medicine, University of Agricultural Sciences and Veterinary Medicine of Cluj-Napoca, 400372 Cluj-Napoca, Romania

**Keywords:** goat milk, mastitis, oxidative stress, pathogens, enzymatic activity

## Abstract

**Simple Summary:**

Demand for goat milk is rising worldwide due to its nutritional characteristics. Inflammation of the mammary gland in goats is one of the most frequently encountered conditions in dairy farms, and it is associated with a decrease in milk quality and changes in milk composition. The aim of this study was to establish parameter levels suggestive of subclinical mastitis by analysing correlations between pathogens and biochemical parameters in goat milk. We collected 76 milk samples (Carpathian goat breed) from one Transylvanian farm in Romania, and we observed that milk from infected mammary glands had a significantly increased somatic cell count and enzymatic activity (lactate dehydrogenase and β-glucuronidase). Milk positive for bacterial growth was associated with oxidative stress, with high concentrations of antioxidant enzymes and oxidation products, as well as oxidative DNA damage. These changes tended to depend on the category of microorganisms isolated from milk samples, some of them being human pathogens, thus posing a threat to public health. According to the present results, assessment of SCC correlated with any of the evaluated biochemical markers, such as inflammatory enzymes, oxidative stress markers and/or oxidative DNA damage indicator, could be used in the early detection of mastitis in farms, especially when important pathogens are involved.

**Abstract:**

Goat mastitis is still frequently diagnosed in dairy farms, with serious consequences on milk quality and composition. The aim of this study was to establish correlations between milk microorganisms and biochemical parameters in goats with no signs of clinical mastitis. Thus, 76 milk samples were collected from a dairy goat farm, Carpathian breed, followed by microbiological, molecular (16S rRNA sequencing) and somatic cells analysis, determination of lactate dehydrogenase (LDH), β-glucuronidase, catalase (CAT), glutathione peroxidase (GPx) activity, total antioxidant capacity (TAC), nitric oxide (NO) and lipid peroxides (LPO) using spectrophotometry and the ELISA method for 8-hydroxy-deoxyguanosine (8-OHdG) as the oxidative DNA damage indicator. Samples positive for bacterial growth showed a significant (*p* < 0.05) increase in the number of somatic cells, LDH and β-glucuronidase activity, as well as higher levels of CAT, GPx, NO, LPO and 8-OHdG compared with pathogen-free milk whereas TAC was lower in milk from an infected udder. These findings suggest that subclinical mastitis is associated with increased enzymatic activity and induction of oxidative stress. Nevertheless, changes in biochemical parameters tended to vary depending on the pathogen, the most notable mean values being observed overall in milk positive for *Staphylococcus aureus*.

## 1. Introduction

Milk is one of the most popular and valuable food sources of animal origin worldwide, and the demand for milk and other dairy products is increasing as the world population is growing. According to the Food and Agriculture Organization (FAO), it is predicted that by 2050 the consumption of animal-origin proteins, including milk and dairy products, will increase by 17% in the Business as Usual (BAU) scenario and 20.9% per capita in the Stratified Societies (SS) scenario [1]. In recent years, goat milk has gained a great deal of attention among consumers due to its higher digestibility, higher content of medium-chain triglycerides, oligosaccharide, β-casein and lower αS1-casein, resulting in reduced allergenicity compared with cow milk [2,3,4,5,6,7,8].

Among the diseases that are frequently diagnosed in a small ruminant farm, mastitis is one of the most encountered conditions that affect the health status of animals [9]. Mastitis is defined as the inflammation of the mammary gland, which mainly occurs after an intramammary infection with different pathogens, such as bacteria, fungi or viruses, but physical, chemical or mechanical injuries can lead to the inflammation of the udder [10]. Unfortunately, mastitis is responsible for major economic losses worldwide due to expensive veterinary treatments, a decrease in milk production and even animal culling [11,12]. Furthermore, mastitis is usually associated with a decrease in milk quality and changes in overall milk composition, and this disease can pose a serious threat to public health due to the fact that raw milk and products from raw milk containing pathogenic microorganisms can enter the food chain and be responsible for food poisoning episodes among consumers [13,14].

Regarding bacterial mastitis, the most frequently isolated pathogens from goat milk are *Staphylococcus* spp. and *Streptococcus* spp. [15,16]. In goats, subclinical mastitis is thought to have a higher prevalence compared with clinical forms. Therefore, the prevalence ranges between 5% and 30% or higher [17,18], whereas clinical mastitis prevalence is lower than 5% [13,17]. Regarding subclinical mastitis etiology, studies have shown that non-aureus staphylococci (NAS) are the most isolated bacteria, whereas *Staphylococcus aureus* bacteria belonging to Enterobacteriaceae family, *Trueperella pyogenes*, *Mycoplasma* spp., *Pseudomonas aeruginosa* or *Bacillus cereus*, are mainly involved in causing clinical forms of mastitis [10,17,19].

One of the most important consequences of mastitis is numerous changes in milk composition. These changes can vary according to the severity of the lesions in the mammary glands and also the degree of damage to the parenchyma [20]. Nonetheless, pathogens involved in the etiology of mastitis play a crucial role in this process, as these changes that occur in mastitic milk also depend on the microorganisms involved [20]. Some of the changes that have been observed regarding goat mastitis are a decrease in milk yield, a reduction in protein and fat content and also an increase in whey proteins, as well as chloride and sodium content [21]. Milk indigenous enzymes tend to increase during mastitis, and, on the other hand, enzymes involved in milk synthesis have lower levels when compared with a healthy udder [22].

Inflammation is accompanied by the presence of polymorphonuclear cells (PMN), which, among others, secrete hydrolytic enzymes that are classified into two major categories: lysosomal, such as N-acetyl-β-glucosaminidase (NAG), β-galactosidase, α-mannosidase, β-glucuronidase and also non-lysosomal enzymes, for example, lactate dehydrogenase (LDH) [23,24]. The latter is a ubiquitous enzyme present at the cytoplasm level of animal cells, and its activity has been shown to increase during mastitis according to the results of research mainly conducted on bovine milk [25,26]. Thus LDH can be considered a reliable indicator of mastitis [27,28]. On the other hand, one of the lysosomal enzymes that is thought to be a suitable marker of mastitis is β-glucuronidase; this molecule is one of the most selectively secreted enzymes during inflammation [23].

Furthermore, inflammation of the mammary gland in ruminant species is accompanied by a rise in the somatic cell count (SCC). This parameter is considered a useful tool, especially in establishing a diagnosis of subclinical mastitis in large ruminants [29,30]. Unlike cow milk, where SCC exceeding 200,000 cells/mL is considered mastitic milk according to the guidelines for bovine milk formulated by the International Dairy Federation (IDF) [31], no generally accepted threshold has been set for goat milk. This is due to the goat’s mammary gland physiology and mechanisms of lactogenesis. In this species, milk secretion is apocrine; thus, numerous cytoplasmic particles with origins in the apical portion of the secretory cells are normally present in the milk, leading to a false increase of SCC because cytoplasmic particles are similar in size to somatic cells. Therefore, DNA-specific staining methods (flow cytometry) are necessary in order to differentiate between these particles and somatic cells [32]. The high percentage of PMN (45–74%) from the total somatic cells in mastitis-free goats suggests that these animals could be more resistant to intramammary infections when compared with cows or sheep [33]. Nonetheless, besides the presence of an infection in the udder, there are also several other non-infectious factors that influence SCC in goats [34,35]. Among these factors are the stage of lactation, parity, breed, milking time, type of milking, feed, farming system, seasonality or stress [29,34,36,37,38].

In recent years, various authors have tried to establish breakpoint values for SCC that would differentiate between a healthy and an infected mammary gland in goats. Most papers focused on goat milk have suggested an interval of 500 to 1000 × 10^3^ cells/mL [39,40]. However, the need for establishing a standard regarding SCC in goat milk is still present [41]. In addition, the sensitivity (Se) and specificity (Sp) of SCC have been investigated in several studies in order to estimate test properties and its use as a screening tool to detect intramammary infection in goats, using different selected cut-offs [38,42,43]. However, the prevalence of infection, lactation stage and parity are all considered factors that influence test performance [42]. Moreover, when analysing somatic cells, many researchers choose to transform SCC into log SCC when data is not normally distributed for a more relevant statistical analysis [42,44].

Besides changes in milk composition, mastitis is also associated with oxidative stress, as studies on ruminant milk show [45,46]. As a definition, oxidative stress is an imbalance between high levels of reactive oxygen species (ROS) and low activity of antioxidant mechanisms of the body [47]. Proteins, lipids and nucleic acids represent the main targets of ROS, as these biomolecules are prone to oxidation [46]. During inflammatory processes in the mammary gland, an accentuated lipid peroxidation can be observed, which leads to a decrease in molecules with an antioxidant role, with the consequent induction of oxidative stress [48]. In order to prevent the over-accumulation of ROS, superior organisms have developed enzymatic and non-enzymatic antioxidant mechanisms. Antioxidant enzymes are catalase, superoxide dismutase, peroxidases, etc., whereas non-enzymatic antioxidants are vitamins A, E, C, carotenoid pigments, ubiquinones, glutathione, cysteine and uric acid, as well as some amino acids or selenium [49,50].

Milk has its own antioxidant mechanism that prevents oxidation processes. Hence, enzymes such ascatalase (CAT), glutathione-peroxidase (GPx), lactoperoxidase or xanthine oxidase play an important role in maintaining milk processing quality, as well as other compounds, such as vitamins A, E and C [51,52]. Studies have shown that during mastitis, the activity of antioxidant enzymes in milk increases, but most of the studies are focused on cow milk, whereas data regarding goat milk and oxidative stress is limited or focuses only on one antioxidant enzyme, such as lactoperoxidase [53] or nitric oxide [54].

Apart from oxidative stress, the concept of nitrosative stress has been gaining attention in recent years. Nitrosative stress is associated with the accumulation of reactive nitrogen species (RNS), such as nitric oxide (NO) and its metabolites [55]. NO plays an important role in the udder’s immune system due to its bactericidal and bacteriostatic activity and is demonstrated to have cyclical activity in milk [56]. Intramammary infections are associated with a higher level of NO in milk [46].

One of the most widely used biomarkers for measuring oxidative stress is 8-hydroxydeoxyguanosine (8-OHdG), an oxidised DNA nucleoside resulting from DNA damage by ROS [57,58]. The concentration of this product can be measured by means of immunohistochemistry, ELISA or chromatographic methods, such as HPLC or LC-MS/MS [58]. Over time, various inflammatory and degenerative diseases have been associated with higher levels of 8-OHdG, as studies performed on humans have shown [58]. Most commonly, the measurement of this compound is performed from serum or urine, but the concentration of 8-OHdG has also been determined from breast milk [59]. However, to the best of our knowledge, no studies on the determination of this parameter from goat milk have been published yet.

The aim of this study was to establish parameter levels suggestive of subclinical mastitis by analysing the correlations between pathogens and biochemical parameters in goat composite milk. In order to achieve this, the following objectives were addressed: (1) assessing the microbiological flora of milk collected from goats without signs of clinical mastitis; (2) evaluation of SCC, LDH and β-glucuronidase as inflammation markers; and (3) analysis of oxidative and nitrosative stress markers and oxidative DNA damage.

## 2. Materials and Methods

### 2.1. Sample Collection

Milk samples (*n* = 76) were collected during spring morning milking from lactating Carpathian goats on a private farm in the Transylvania region of Romania. The average weight of the females was 46.38 kg, the average age was 1.7 years (second lactation), with a 240–280 L/lactation period (approximately 9 months), and the fertility rate was 1.5. General mean physico-chemical parameters at the population level were lactose 4.7%, protein 3.44% and fat 4.29% (unpublished data from the farm’s milk quality monitoring program). Animals included in the study were in their 4th to 14th week of lactation. Females under the first month of lactation were excluded. Generally, goats were managed under the traditional system based on pasture (semi-intensive farming) with access to pasture grazing during daylight, from spring to autumn. In the wintertime, animals were kept in stalls, and the winter diet was a mixture of hay, lucerne (*Medicago sativa*) and corn (*Zea mays*). According to the farm veterinarian, the cumulated incidence of clinical mastitis in the last 5 years reached 5% of the population (*n* = 187). Animal welfare, sanitary and hygienic aspects were respected, and water was available ad libitum. Goats were milked twice a day, in the morning and in the evening using manual milking.

All animals included in the study were subjected to a clinical evaluation in order to exclude conditions that would influence general health status. In addition, an udder inspection was performed in order to exclude females with mastitis. Thus, inspection and palpation of the mammary gland were made in search of different macroscopic signs of clinical inflammation (hyperemia, presence of scabs, nodules, ulcers, increased local temperature and sensitivity, reactive supramammary lymph nodes and changes in udder texture). Goats presenting clinical signs of mastitis were excluded, as well as animals receiving antibiotic treatments. In addition, milk organoleptic analysis was performed using approximately 15–20 mL of milk (the first 3–4 squirts of milk) in order to detect any sensorial change regarding aspect, colour or smell. Thus, milk samples were visually inspected for signs of defect, such as curdling, presence of blood or abnormal colour, followed by the assessment of smell, paying attention to any abnormal odour. All the animals and samples were examined by the same person.

Following the cleaning of the udder of each goat included in the study with water and soap, proper disinfection of the teat ends was performed with 70% alcohol, followed by the withdrawal of the first 3–4 squirts of milk, used for organoleptic analysis and subsequently discarded. Next, sterile recipients with caps were used to collect 50 mL from each udder half, gathering a total of 152 udder halves from 76 animals. Milk samples were processed as composite samples. Each goat was sampled once, all on the same day. Milk was then stored in isothermal containers (4–8 °C) and transferred to the laboratory on the same day for further processing and bacteriological and biochemical testing.

### 2.2. Microbiological Analysis

Upon arrival at the microbiology laboratory, 10 µL of milk was used to streak Columbia Agar with 5% sheep blood (Biomaxima, Lublin, Poland) and also MacConkey Agar (Oxoid Ltd., Hampshire, UK) plates and were then incubated at 37 °C for 24 h. If no bacterial growth was detected after 24 h of incubation, then the plates were reincubated for another 24 h. After incubation, all plates were examined for bacterial growth, and a preliminary identification was performed by evaluating colony morphology, the presence or absence of hemolysis and tinctorial and morphological features of the cells following the Gram staining. A slide catalase test with 3% hydrogen peroxide and an oxidase test using strips were performed [16]. Milk samples were considered microbiologically positive when they yielded at least one bacterial colony. Samples yielding more than two types of colonies were considered contaminated. If samples showed no bacterial growth after 48 h of incubation, they were considered negative [31].

#### Bacterial Species Identification in Milk

Isolated colonies from each sample were used to inoculate Columbia Agar with 5% sheep blood (Biomaxima, Lublin, Poland) by the mechanical streaking method and incubated for 24 h at 37 °C in order to obtain pure cultures for bacterial species identification by 16S rRNA gene sequencing. Identified bacterial isolates were preserved in 60% glycerol broth at −20 °C.

DNA extraction

The bacterial DNA extraction was performed from each colony type using Chelex^®^ 100 Resin Molecular Biology Grade Resin (Bio-Rad, Hercules, CA, USA) according to the protocol described by Dan et al. in 2015 [60]. Briefly, 150 µL of Chelex^®^ 100 (10%) were added to approximately 2–5 bacterial colonies. The 150 µL of Chelex^®^ 100 was previously sterilised for 30 min in Eppendorf tubes under UV in a microbiological flow class II. The aliquots were briefly mixed and incubated at 57 °C for 30 min, then incubated for a further 5 min at 94 °C. After 14,000 RPM for 1 min, the supernatant was collected and transferred to another sterile 1.5 mL tube and was used in the PCR reaction. Molecular analysis was performed on each colony type isolated from the 66 goat milk samples. The purity of the DNA was measured as the ratio of absorbance at 260/280 nm. The ratio (purity) varied between 1.73–1.8 and between 50–99 ng/ul for the concentration. The concentration and purity of the DNA extracts were evaluated in a representative number of samples through a random procedure using the Nanodrop ND-1000 Spectrophotometer (NanoDrop Technologies, Inc., Wilmington, DE, USA).

PCR and gel electrophoresis

The PCR of 16S rRNA gene was performed in a G-Storm GS1 Thermal Cycler (Cambridge Scientific, Watertown, MA, USA) using 2x Red PCR Master Mix (RovalabGmBH, Teltow, Germany) in a final volume of 25 µL as follows: 12.5 µL of Master Mix, 1 µL (10 pmol) of each primer (27F: AGAGTTTGATCCTGGCTCAG, 1492R: TACGGYTACCTTGTTACGACTT), 4 µL of water and 6.5 µL of extracted DNA from each sample. The PCR conditions consisted of 3 min at 96 °C, 30 cycles of 20 s at 94 °C, 40 s at 58 °C, 40 s at 72 °C and a final extension at 72 °C [61]. Negative and positive controls were used for quality control of the PCR. The expected product size was around 1300–1400 bp.

Agarose gel (1.5%) electrophoresis (Cleaver Scientific Ltd., multiSUB™ Midi, Rugby, UK), stained with SYBR Safe DNA gel stain (Invitrogen, Carlsbad, CA, USA), was performed for the visualization of PCR products.

DNA sequencing

All the PCR-positive products were purified by using ISOLATE II PCR and Gel Kit (Bioline, Meridian, UK) and sequenced at Macrogen Inc. (Amsterdam, The Netherlands). Nucleotide sequences were compared with those available in GenBank using Basic Local Alignment Search Tool (BLAST) analysis.

### 2.3. SCC Assessment

The SCC evaluation was performed on fresh milk on the same day as the sample collection using the Lactoscan Somatic Cell Counter (Milkotronic Ltd., Nova Zagora, Bulgaria). This system is based on a method of counting individual cells by detecting the fluorescent signals of DNA in the cell nucleus. The assessment was performed according to the manufacturer’s instructions.

### 2.4. Enzymatic Activity Assessment

In order to conduct biochemical determinations, immediately after microbiological testing, milk samples were divided and processed as follows: whole milk from each sample was added into Eppendorf tubes and stored in a deep freezer (−80 °C) while the leftover quantity was used to obtain skimmed milk after centrifugation at 4200 RPM for 10 min in a cooling centrifuge at 4 °C (Biosan, Riga, Latvia). The fat layer in each sample was removed, and the defatted milk was immediately stored at −80 °C until further analysis. Whey proteins were separated from caseins by isoelectric precipitation from skimmed milk with 10% acetic acid at pH 4.1, followed by centrifugation at 4000 RPM and 5 °C for 15 min [62]. Lactoserum was the clear supernatant, while the sediment represented the casein fraction. The obtained lactoserum was distributed in Eppendorf tubes and stored (−80 °C) until biochemical analysis was performed.

Lactate dehydrogenase (LDH)

LDH activity was assessed in skimmed goat milk using the commercial kit LDH-P Stable Liquid (AMP Diagnostics, Graz, Austria), which is a kinetic enzymatic method. The activity of this enzyme can be measured either through the transformation of lactate into pyruvate or through the reverse reaction of pyruvate into lactate. The kit used is based on the second reaction in which LDH catalyzes the oxidation of pyruvate into lactate with the simultaneous oxidation of NADH into NAD. Thus, the rate of NADH oxidation can be measured as a decrease in absorbance, and this rate is directly proportional to the LDH activity in the tested sample. Milk samples were processed according to the kit instructions and were analysed using a Screen Master Touch analyser (Hospitex Diagnostics, Florence, Italy) at a wavelength of 340 nm. Results were expressed in U/L.

β-glucuronidase

The activity of this enzyme was evaluated using a previously described protocol [23]. For the synthetic enzymatic substrate, 4-nitro-phenyl β-D-glucuronide pnPG (PanReac AppliChem, ITW Reagents, Darmstadt, Germany) was used. Briefly, the assay was performed with 0.4 mL skimmed milk, 0.2 mL pnPG 40 mM and 0.4 mL 1 M acetate buffer, followed by a 4-h incubation at 50 °C. The next step was adding 0.5 M carbonate buffer 4 mL as a stop solution, followed by centrifugation of 3000× *g* for 20 min. The p-nitrophenol from the supernatant liberated by the synthetic substrate was measured at a wavelength of 410 nm using a spectrophotometer (Spectrostar Nano, BMG Labtech, Ortenberg, Germany). The results were calculated using a standard curve [23] and expressed in units (U).

### 2.5. Oxidative Stress Markers Evaluation

Catalase activity

Determination of catalase activity in skimmed milk samples was performed using the commercial Catalase (CAT) Assay Kit (Elabscience Biotechnology Inc., Houston, TX, USA) based on the colorimetric method. The principle behind this kit is the following: the reaction in which catalase breaks down hydrogen peroxide (H_2_O_2_) is stopped by ammonium molybdate. The residual hydrogen peroxide will react with the ammonium molybdate, generating a yellowish complex whose absorbance is evaluated at a wavelength of 405 nm. The optical densities were evaluated using a spectrophotometer (Spectrostar Nano, BMG Labtech, Ortenberg, Germany) against distilled water in 0.5 cm cuvettes. The amount of catalase in a 1 mL sample that decomposes 1 μmol H_2_O_2_ per minute at 37 °C is defined as one unit.

Glutathione peroxidase activity

The commercial Glutathione Peroxidase (GSH-Px) Assay Kit (Elabscience Biotechnology Inc., Houston, TX, USA), colorimetric method, was used to evaluate glutathione peroxidase activity in skimmed goat milk samples, following the instructions of the manufacturer. In the end, the absorbance of each sample was measured against distilled water at 412 nm (Spectrostar Nano, BMG Labtech, Ortenberg, Germany) using 1 cm cuvettes.

Lipid peroxides quantification

The commercial Lipid Peroxidation LPO Assay Kit (Shanghai Coon Koon Biotech Co., Ltd., Shanghai, China) was used to evaluate this parameter. Analysis was performed on skimmed milk samples according to the producer’s instructions. The absorbance of each sample was measured spectrophotometrically (Spectrostar Nano, BMG Labtech, Ortenberg, Germany) at a wavelength of 586 nm, and results were expressed in µmol/L.

Determination of milk’s total antioxidant capacity

Assessment of the total antioxidant capacity of skimmed milk samples was performed using the commercial Total Antioxidant Capacity (T-AOC) Colorimetric Assay Kit (Elabscience Biotechnology Inc., Houston, TX, USA) following the instructions supplied by the kit.

Determination of nitric oxide levels in milk

In order to evaluate NO levels in the goat milk samples included in the study, a commercial kit was used, Nitric Oxide (NO) Colorimetric Assay Kit (Elabscience Biotechnology Inc., Houston, TX, USA), and the analysis was performed on goat milk lactoserum. NO can easily oxidise in aqueous solutions and form ${\mathrm{NO}}_{2}^{-}$, and a reddish compound is formed following the reaction with the chromogenic agent. The concentration of this compound is proportional to NO concentration, which was indirectly calculated by measuring the absorbance at 550 nm using 1 cm cuvettes (Spectrostar Nano, BMG Labtech, Ortenberg, Germany). In the end, the obtained results were expressed in µmol/L.

Determination of DNA damage using 8-hydroxy-deoxyguanosine assessment

The 8-hydroxy-deoxyguanosine (8-OHdG) concentration was determined using a commercial ELISA kit (Elabscience Biotechnology Inc., Houston, TX, USA), which uses the Competitive-ELISA principle. For this assay, skimmed milk samples were used, and the producer’s protocol was followed. The optical density was spectrophotometrically measured at a wavelength of 450 nm using a microplate reader (Spectrostar Nano, BMG Labtech, Ortenberg, Germany). The concentration of 8-OHdG was calculated by comparing the optical density of the samples with the standard curve, and the results were expressed in ng/mL.

### 2.6. Statistical Analysis

Statistical analysis was performed using Epi Info 7 software (CDC, Atlanta, GA, USA) and Microsoft Excel functions. For SCC, the normality test Jarque Bera (JB) was performed in order to evaluate data distribution. The remaining parameters were evaluated using Bartlett’s test, and, depending on the *p*-value, they were further evaluated using an ANOVA (parametric) or Mann-Whitney test (non-parametric). The SCC was log10 transformed as it was not normally distributed. The results are presented as means and standard deviation. Correlations between different parameters were calculated using Pearson coefficient (r), and the results for all tests were considered significant at *p* ≤ 0.05.

## 3. Results

### 3.1. Microbiological and Molecular Analysis

Bacteriological analysis was performed on 76 goat milk samples, as previously described, out of which 13.16% (*n* = 10, 95% CI 6.49–22.87) were negative and 66 were positive for bacterial growth, among which 53 (69.74%, 95% CI 58.13–79.74) were single infections and 13 (17.11%, 95% CI 9.43–24.47) with two bacterial species. Following bacterial species identification by 16S rRNA gene sequencing, a total of 27 species were identified, belonging to eight genera: *Staphylococcus*, *Bacillus*, *Enterococcus*, *Aerococcus*, *Streptococcus*, *Moraxella*, *Macrococcus* and *Aeromonas*. According to the number of isolates, *Staphylococcus* was the most frequently isolated genus with 32 isolates, followed by *Enterococcus* (*n* = 24) and *Bacillus* (*n* = 18), whereas a smaller number of isolates belonged to *Macrococcus* (*n* = 4), *Aerococcus* (*n* = 2), *Moraxella* (*n* = 2), *Streptococcus* (*n* = 1) and *Aeromonas* (*n* = 1). According to the number of bacterial species, *Staphylococcus* was the most diverse isolated genus with 13 different species, followed by *Bacillus* (*n* = 6), *Enterococcus* (*n* = 3), *Aerococcus* (*n* = 1), *Macrococcus* (*n* = 1), *Streptococcus* (*n* = 1), *Moraxella* (*n* = 1) and *Aeromonas* (*n* = 1). In the majority of cases, minor pathogens were identified, followed by major ones (*S. aureus*). The 16SrRNA sequences showed 99–100% similarities with the bacterial species presented in Table 1.

Taking into account the bacteriological analysis results, for a better presentation and easier data interpretation, milk samples were grouped into five different categories according to the most frequently isolated bacteria as follows: non-aureus staphylococci group (NAS), milk samples positive for enterococci (E), samples positive for *Bacillus* genus (B), goat milk samples from which *Staphylococcus aureus* was isolated (SA), and a category for microorganisms with the lowest prevalence, grouped under the name “other pathogens” (O). A group represented by microbiologically negative samples (N) was added. The most numerous group was NAS (27.66%), followed by E (25.53%), B (19.15%), N (10.64%), O (10.64%) and SA (6.38%). Furthermore, the NAS, E and B categories were referred to as minor pathogens, whereas the SA category was referred to as major pathogens. Results regarding the O group were not comparatively discussed due to the fact that this particular category was not homogenous, as it comprised both Gram-positive and -negative microorganisms.

### 3.2. SCC

Following the SCC assessment using the Lactoscan Somatic Cell Counter, the results, which are presented in Table 2, were obtained. Microbiologically negative samples were characterised by lower SCC compared with milk samples positive for bacterial growth. Moreover, differences have been observed between sample categories (Table 2).

### 3.3. Enzymatic Activity

Significant differences (*p* ≤ 0.05) were observed between LDH and β-glucuronidase in the categories of the infection status of goat milk, with lower enzymatic activity in the negative sample group for both enzymes compared with samples associated with minor and/or major pathogens. When comparing milk samples positive for bacterial growth, samples associated with *Staphylococcus aureus* exhibited the highest enzymatic activity for both LDH and β-glucuronidase compared with all the other categories (*p* ≤ 0.05). The detailed results for all categories are presented in Table 3.

### 3.4. Oxidative Stress Markers and 8-Hydroxy-Deoxyguanosine Assessment

Following the biochemical determinations using the previously described methods, the obtained results are presented in Table 4 as means and standard deviation, as well as the significant difference among the categories (*p* ≤ 0.05).

Correlations were evaluated using Pearson Correlation Coefficient. When analysing data regarding SCC and LDH and β-glucuronidase activity results, moderate positive correlations were established between SCC and LDH levels (*r* = 0.5105, *p* < 0.05), as well as between SCC and β-glucuronidase (*r* = 0.5404, *p* < 0.05). A weaker positive correlation was observed between these two enzymes (*r* = 0.4295, *p* < 0.05).

Moderate positive correlations were also established between SCC and the analysed oxidative stress markers. Therefore, an increased SCC tended to be associated with increased CAT (*r* = 0.596, *p* = 0.009) and GPx (*r* = 0.4805, *p* < 0.001) activity. Furthermore, NO, LPO and 8-OHdG were positively correlated with SCC showing a moderate association (*r* = 0.4803, *p* = 0.001*; r* = 0.6439, *p* < 0.001; *r* = 0.4687, *p* = 0.002).

No significant correlations were established whatsoever between the number of somatic cells and milk TAC or between LDH and TAC and β-glucuronidase and TAC. A moderate positive correlation was also noted for LDH and GPx (*r* = 0.564, *p* < 0.001), as well as for β-glucuronidase and GPx (*r* = 0.5296, *p* < 0.001), whereas CAT activity was shown to have a weaker positive correlation with LDH (*r* = 0.3435, *p* = 0.002) and β-glucuronidase (*r* = 0.2299, *p* = 0.04) compared with GPx.

## 4. Discussion

The microbiological analysis revealed a high number of positive samples (*n* = 66); thus, the prevalence of subclinical mastitis among the analysed animals was higher compared with other studies, where the prevalence was reported to range between 5% and 30% or higher [17,63]. Nevertheless, the present study’s results showed a high prevalence of non-aureus staphylococci, which is similar to previous papers on subclinical mastitis in goats [27,64,65]. NAS is considered to be a group of pathogens that are easily transferred between hosts [63]. These microorganisms are also reported in other regions as the most frequent cause of intramammary infections [20,66]. Furthermore, some NAS species are known to produce biofilm and therefore contaminate milk, posing a serious threat to public health [67]. To the best of our knowledge, among all species of *Staphylococcus* that were isolated in this study, *S. petrasii* subsp. *jettensis* has not been previously reported in goat milk. *Staphylococcus petrasii* ssp. *jettensis* was isolated in two composite samples (2.63%), being the only isolate from each sample, with 22 and 20 CFU/10 µL milk, respectively. Both goats were in their second lactation, and the SCC was similar (309 × 10^3^ and 372 × 10^3^ cells/mL). This species is thought to be an opportunistic human pathogen with clinical importance due to the presence of virulence factors and multi-drug resistance, including methicillin resistance [68].

*S. aureus* was identified in 7.89% of the positive samples; this result is similar to data reported in other studies [69,70], whereas other results revealed a higher prevalence of *S. aureus* (37%) from goat subclinical mastitis cases in other regions of the world, such as Brazil [66]. Due to the fact that *S. aureus* is involved in the etiology of both clinical and subclinical mastitis, it is considered one of the most important caprine mastitis pathogens, as it is also one of the reasons for animal culling and therapy failures due to antibiotic-resistant strains [66].

Another group of bacteria frequently isolated from goat milk samples was *Enterococcus* spp. (25.53%). Enterococci have been previously isolated from goat milk and goat dairy products [71,72,73,74], and they represent an important part of milk microbiota as lactic acid bacteria (LAB) [73]. LAB are known to produce antimicrobial substances, such as bacteriocins; however, some *Enterococcus* strains can possess virulence genes and antibiotic resistance, thus posing a concern for human health [72]. These strains belong to species such as *E. durans*, *E. faecalis* and *E. faecium*, which were also reported in this study.

The results of this study also highlighted the presence of bacteria belonging to *Bacillus* genus. As ubiquitous microorganisms in the soil, water, agricultural products and manure, they are considered food contaminants [75]. These microorganisms can reach high numbers after contamination during milk collection in farms and milk processing units [76]. However, sporulated strains of *Bacillus* that produce heat-stable toxins pose a risk for dairy products, as both endospores and toxins can survive thermal treatments, such as pasteurization [77]. Regarding *B. licheniformis*, the most frequently isolated *Bacillus* species in the present study, research conducted by Nieminen et al. in 2007 [77] demonstrated the presence of toxin-producing *B. licheniformis* in cow milk, along with *B. cereus* and *B. pumilus*, pulling a warning signal about the potential danger of milk entering the food chain. Both *B. cereus* and *B. pumilus* were isolated in the present study. Although *B. licheniformis* is generally regarded as a non-pathogenic species, its presence has been associated with food poisoning, infant mortality and abortion in cattle [77,78].

Furthermore, *Aerococcus viridans* was also detected in the goat milk samples in our study. This species was isolated in two samples (2.63%), and both cases involved a single infection with 12 CFU/10 µL and 17 CFU/10 µL milk, respectively. This bacterium is considered an emerging pathogen in human medicine due to the fact that it is involved in cases of septicemia, endorcarditis and meningitis [67]. Recently, research conducted by Aragao et al. in 2021 [67] in Brazil reported this microorganism as a causative agent of goat mastitis.

In contrast to other studies in the literature [16,79], the bacteriological analysis of samples included in this study did not reveal microorganisms belonging to *Corynebacterium* genus.

In general, when performing the interpretation of the results obtained after the microbiological analysis of samples, some important aspects need to be taken into account, such as the milk microbiota and the role of commensal and opportunistic bacteria in the udder because, for instance, species that can be isolated from the milk of healthy animals [80] can also be involved in the etiology as mastitis [11], as is the case in *Staphylococcus epidermidis* [80].

When analysing results concerning SCC, significant differences (*p* ≤ 0.05) were observed between the sample categories. Thus, in this study, goat milk was affected by infectious status, with microbiologically positive samples having a higher SCC compared with pathogen-free milk, but significant differences were only observed when comparing N with the NAS, B and SA groups. In addition, a variation in SCC was noted for the E, B, NAS and SA categories, suggesting that different microorganisms lead to an increase in the number of somatic cells but to a smaller or greater extent depending on the pathogenicity of bacteria. Therefore, the enterococci group (E) showed the lowest mean for SCC among all bacteriologically positive samples, with average values very similar to the negative (N) sample group, suggesting that bacteria belonging to *Enterococcus* genus do not lead to a significant increase in SCC values. This could be explained by the fact that enterococci are commonly found in raw goat milk, as a part of a dominant flora representing LAB, along with *Lactobacillus*, *Leuconostoc* or *Lactococcus* [80].

Nevertheless, higher mean values for SCC were observed in milk samples belonging to the B and NAS groups, with rather similar mean values but a higher count whatsoever when compared with negative samples. By far the highest SCC was observed for the SA group, with a mean value of 4377.83 × 10^3^ cells/mL. This finding is consistent with the results of other studies that obtained a very high count in major pathogens infection [81]. However, other studies have detected similar changes regarding SCC in goat milk positive for NAS and major pathogens [82]. Moreover, the fact that NAS is a very heterogeneous group could be a possible explanation for the variations in SCC observed in this study regarding NAS-infected samples; thus, further investigations are needed at the species level in order to have a detailed picture of how this bacterial group truly influences the number of milk somatic cells in goats.

Many authors have attempted to establish a cut-off value for SCC in goats, but data are still not consistent, although efforts are being made. However, at this time, there is no legislation in the EU for goat milk somatic cells. In contrast, the United States legislation established a maximum of 1,500,000 cells/mL for bulk-tank goat milk [83].

Results concerning milk enzymatic activity have shown that both LDH and β-glucuronidase significantly increased activity in microbiologically positive samples (all categories) in comparison with the negative group. The results of the present study are in line with other studies conducted on small ruminants (goats and sheep) that have demonstrated that milk’s LDH levels in subclinical mastitis are higher compared with healthy milk [28,84]. Some authors suggest that LDH is one of the most reliable markers for the detection of mastitis, along with two other enzymes, alkaline phosphatase and aspartate aminotransferase [28]. Moreover, similar results have also been reported for cow milk, with results showing that LDH activity was higher during mastitis [24,25,85] and that enzymatic levels were different depending on the isolated pathogen [82]. In this study, the authors suggested that intramammary infection caused by Gram-negative bacteria leads to a higher LDH level compared with a Gram-positive infection [85]. In the present study, the highest mean value was observed in the SA group, with statistically significant differences (*p* ≤ 0.05) observed for the SA and NAS, E and B groups, respectively. Even though in milk samples belonging to the E group the average SCC was low, LDH concentration was higher compared with the negative group (*p* ≤ 0.05), suggesting that not only white blood cells and somatic milk cells secrete this enzyme but also invading microorganisms [24]. The positive association between the number of somatic cells and LDH enzymatic activity suggesting that a high SCC goes with increased milk enzymatic levels is not a surprising finding, taking into account that high levels of LDH are associated with mammary tissue destruction during intramammary infection [24,85].

Furthermore, results regarding β-glucuronidase levels in goat milk samples showed significantly lower activity in healthy milk in comparison with bacteriologically positive samples; this finding is consistent with other studies’ results on goat milk [23,27]. In addition, results showed that *S. aureus* infection is associated with the highest enzymatic activity, followed by the B and NAS groups with similar mean values and the E group. The obtained data suggest that even in cases of subclinical mastitis in goats, major pathogen infections lead to severe injuries to the udder tissue, which may be associated with inflammation and increased milk enzymatic activity. Since statistically significant differences were observed between all categories, β-glucuronidase could represent a reliable indicator of subclinical mastitis caused by either major or minor pathogens.

Results concerning antioxidant enzymes show that milk from healthy animals with a negative bacteriological test had significantly lower activity compared with milk samples positive for bacteria. Thus, regarding CAT, negative samples showed a decreased activity compared with the other categories, especially the SA category. A similar trend was reported by Silanikove et al. [54]. The same observation is available for GPx, where the major pathogen group (SA), followed by minor pathogens (the NAS and B groups), are associated with significantly increased enzymatic activity compared with microbiologically negative milk samples. CAT plays an essential role in maintaining cellular redox balance by eliminating hydrogen peroxide and transforming it into water and oxygen and so does GPx as a membrane protector against oxidative damage by inhibiting lipid peroxidation [86].

Previous research has shown that milk from animals diagnosed with subclinical mastitis was characterised by an increase in GPx activity compared with healthy milk [87], and the results of the present study on goat milk are consistent with previous papers [88]. The increased enzymatic activity could be explained by both the hydrolysis of the casein-enzyme complex, which is followed by enzyme release, or by the possible pathogen’s antioxidant defense system as a survival mechanism [52]. The origin of these antioxidant enzymes may be represented not only by blood but also by milk fat globule membranes and somatic cells, which could explain the increased enzymatic activity in milk with high SCC. This observation is also supported by the positive correlation between both SCC and GPx (*r* = 0.4805, *p* < 0.0001) and SCC and CAT (*r* = 0.596, *p* = 0.009).

When analysing data related to milk total antioxidant capacity (TAC), these have shown that this parameter presents significant changes, with lower mean values in milk samples positive for bacteria, especially in *S. aureus* infection, when compared with healthy milk. However, for the SA group significant differences were only observed in comparison with the N and B groups. These results are in line with other studies on goats [54] and cows [45,89]. The results of the present study suggest that goat milk’s antioxidant status may be influenced by the presence of pathogens and, most importantly, by the severity of infection and the type of microorganism.

Data obtained for lipid peroxidation showed that milk samples included in the NAS, E, B and SA categories had a significantly higher concentration of lipid peroxide compared with the negative group, and, as was expected, the SA group expressed the highest mean values. The positive correlation between SCC and LPO indicates that the more severe the inflammatory process in the udder, the higher the level of lipid peroxidation. These results are consistent with other studies on mastitis in cow milk [45] that demonstrated that oxidative degradation of lipids is accentuated in case of intramammary infection.

As previously reported, inflammation is associated with NO production as it is activated by cytokines [48]. Both udder epithelial cells and macrophages secrete NO, thus explaining the high amounts found in the present study of milk samples associated with bacterial infection that are significantly different from the N group. These findings are in line with other studies’ results [54,89]. It is known that during inflammation (e.g., mastitis) NO reacts with superoxide anion and forms peroxinitrite radicals, which target membrane fatty acids, resulting in an increased lipid peroxidation [48]. This affirmation is emphasised by the positive correlation established between NO and LPO, suggesting that increased NO is associated with high levels of LPO in intramammary infection.

Oxidative stress has negative effects on the organism, one of them being DNA damage during oxidative processes induced by ROS; 8-OHdG is one of the most commonly used markers for the assessment of DNA damage and due to the fact that degenerative and inflammatory diseases have been associated over time with high levels of 8-OHdG [58]. As expected, the results of the present sudy revealed significantly higher concentrations of 8-OHdG in microbiologically positive milk samples (major pathogens followed by minor pathogens) compared with pathogen-free milk, indicating the presence of oxidative stress followed by the oxidative DNA damage in milk from an infected mammary gland. However, there is currently no literature data regarding the assessment of this parameter in goat milk, as the only data regarding milk is associated with breast milk [59,90].

Although the results of the present study show a correlation between the analysed parameters leading to the possibility to use any of these markers in adition to the SCC and/or pathogens’ presence in the evaluation of subclinical mastitis, these should be confirmed by future studies. Due to the fact that the study was aimed to be conducted on a homogeneous population (origin, environmental and physiological characteristics) in order to reduce variables, the resulting sample size was small, constituting an important limitation. Since only goats without clinical signs of mastitis were included, a low bacterial load was expected. This was the reason for deciding to analyse composite milk samples, which can represent a drawback, especially in SCC evaluation. As expected, the number of isolated bacterial species was low, not allowing a statistical analysis on each species. Because of this limitation, samples were grouped depending on the category of microorganism (genus or species). However, this could influence results, since different bacterial species could have different pathogenicity and virulence factors, thus possibly influencing studied parameters, especially inflammatory markers.

This study included a complex approach to goat milk, taking into account several markers (associated with inflammation, oxidative and nitrosative stress) in addition to SCC. Among these, the evaluation of 8-OHdG as an oxidative DNA damage marker was performed, being the first study to have included this parameter in the assessment of milk biochemical characteristics.

In an attempt to find alternative diagnosis methods for subclinical mastitis, this paper aimed to study the correlations among all these milk parameters. Nowadays, there are no official definitions of subclinical mastitis in goats, but according to our results, this condition could be defined as “changes in milk composition, such as increased SCC over 500,000 cells/mL and one of the following biochemical markers changes: increased enzymatic activity, oxidative and nitrosative stress markers, low antioxidant capacity and/or presence of major pathogens, with no signs of inflammation or visible changes in milk”. Altogether, the present results, along with the proposed definition above of goat subclinical mastitis, open new perspectives in the field, inspiring other researchers to use the same approach and, by this, confirm our results. If confirmed, all this could lead to more research in developing alternative diagnostic tools to be used in the farms.

## 5. Conclusions

In conclusion, in the present study, analysed parameters in goat milk suggest that increased SCC, oxidative (CAT, GPx, LPO and 8-OHdG) and nitrosative (NO) stress markers, inflammatory enzymes (LDH and β-glucuronidase) and decreased TAC can be associated with the presence of important pathogens, such as *S. aureus* and present a moderate correlation among each other. Thus, the assessment of SCC, milk enzymatic activity, as well as the evaluation of oxidative and nitrosative markers, could be used in the early detection of mastitis on farms, especially in the case of major pathogen involvement. One of these could be used as an alternative for the current methods depending on the farm choice and resources. However, further research is needed in order to strengthen these observations in more diverse populations, followed by the possible development of rapid tests for these parameters.

To the best of our knowledge, this is the first study in Romania focused on goat milk, in a complex approach, trying to establish parameter levels suggestive of subclinical mastitis by analysing correlations between pathogens and biochemical parameters. Moreover, the present study reports, for the first time, the isolation of *S. petrasii* subsp. *jettensis* from raw goat milk. The novelty of this approach consists in the evaluation of 8-OHdG as an oxidative DNA damage marker in goat milk samples.

## Figures and Tables

**Table 1 animals-12-03245-t001:** The frequency of bacterial species identified by BLAST analysis of 16S rRNA gene sequence in 76 composite mid-lactation milk samples from Carpathian goats without signs of clinical mastitis in Romania.

Species	No. Isolates/Frequency %	CI 95%
*Enterococcus durans*	14/18.42	10.45–28.97
*Bacillus licheniformis*	8/10.53	4.66–19.69
*Enterococcus faecium*	8/9.21	3.78–18.06
*Staphylococcus aureus*	6/7.89	2.95–16.40
*Staphylococcus caprae*	5/6.58	2.17–14.69
*Staphylococcus epidermidis*	5/6.58	2.17–14.69
*Bacillus subtilis*	4/5.26	1.45–12.93
*Macrococcus caseolyticus*	4/5.26	1.45–12.93
*Staphylococcus hominis*	4/5.26	1.45–12.93
*Bacillus pumilus*	3/3.95	0.82–11.11
*Staphylococcus chromogenes*	3/3.95	0.82–11.11
*Aerococcus viridans*	2/2.63	0.32–9.18
*Enterococcus faecalis*	2/2.63	0.32–9.18
*Moraxella osloensis*	2/2.63	0.32–9.18
*Staphylococcus haemolyticus*	2/2.63	0.32–9.18
*Staphylococcus petrasii* subsp. *jetensii*	2/2.63	0.32–9.18
*Aeromonas hydrophila*	1/1.32	0.03–7.11
*Bacillus cereus*	1/1.32	0.03–7.11
*Bacillus clausii*	1/1.32	0.03–7.11
*Bacillus thuringiensis*	1/1.32	0.03–7.11
*Staphylococcus cohnii*	1/1.32	0.03–7.11
*Staphylococcus equorum*	1/1.32	0.03–7.11
*Staphylococcus sciuri*	1/1.32	0.03–7.11
*Staphylococcus vitulinus*	1/1.32	0.03–7.11
*Staphylococcus xylosus*	1/1.32	0.03–7.11
*Streptococcus pseudoporcinus*	1/1.32	0.03–7.11
Total	84 ^1^/86.84 ^2^	77.13–93.51 ^2^

95% CI: 95% confidence interval; ^1^ total number of isolates from 76 goat milk samples; ^2^ frequency with 95% CI of bacteriologically positive samples.

**Table 2 animals-12-03245-t002:** SCC and log-transformed SCC and standard deviation by microbiological category based on BLAST analysis of 16S rRNA gene sequence in 76 composite mid-lactation milk samples from Carpathian goats without signs of clinical mastitis in Romania.

Microbiological Category (n)	SCC (×10^3^ Cells/mL) Mean ± SD	Log_10_ SCC Mean ± SD	Significant Difference from (*p* ≤ 0.05)
N (10)	236.4 ± 64.1	5.36 ± 0.11	NAS, B, SA
NAS (26)	710.52 ± 458.02	5.76 ± 0.27	N, E, SA
E (24)	251.75 ± 112.7	5.36 ± 0.17	NAS, B, SA, O
B (18)	709.83 ± 385.91	5.79 ± 0.22	N, E, SA
SA (6)	4377.83 ± 1426.65	6.62 ± 0.12	N, NAS, E, B, O
O (10)	871.9 ± 1478.12	5.64 ± 0.54	E, SA

Samples categorised according to the identified bacterial genus or species: N—negative samples, NAS—non-aureus staphylococci group, E—enterococci group, B—*Bacillus* spp. group, SA—*S. aureus* group, O—other pathogens group.

**Table 3 animals-12-03245-t003:** Milk LDH and β-glucuronidase activity results (mean ± SD) by microbiological category based on BLAST analysis of 16S rRNA gene sequence in 76 composite mid-lactation milk samples from Carpathian goats without signs of clinical mastitis in Romania.

Enzyme	Microbiological Category					
	N	NAS	E	B	SA	O
LDH (U/L)	125.92 ± 17.47	287.84 ± 81.47	270.92 ± 62.33	253.15 ± 69.7	446.71 ± 23.28	301.02 ± 149.57
Signif. dif. (*p* ≤ 0.05)	NAS, E, B, SA, O	N, SA	N, SA	N, SA	N, NAS, E, B, O	N, SA
β-glucuronidase (U)	19.23 ± 3.51	35.16 ± 10.2	26.29 ± 9.34	39.06 ± 12.84	60.92 ± 3.35	31.46 ± 11.95
Signif. dif. (*p* ≤ 0.05)	NAS, E, B, SA, O	N, E, B, SA, O	N, NAS, B, SA, O	N, NAS, E, SA, O	N, NAS, E, B, O	N, NAS, E, B, SA

Samples categorised according to the identified bacterial genus or species: N—negative samples, NAS—non-aureus staphylococci group, E—enterococci group, B—*Bacillus* spp. group, SA—*S. aureus* group, O—other pathogens group; Signif. dif.: significantly different from.

**Table 4 animals-12-03245-t004:** Oxidative stress markers results (mean ± SD) by microbiological category based on BLAST analysis of 16S rRNA gene sequence in 76 composite mid-lactation milk samples from Carpathian goats without signs of clinical mastitis in Romania.

Parameter	Microbiological Category					
	N	NAS	E	B	SA	O
CAT (U/mL)	1.54 ± 0.24	2.85 ± 1.45	2.63 ± 1.51	2.51 ± 0.56	3.92 ± 0.31	3.37 ± 1.34
Signif. dif. (*p* ≤ 0.05)	NAS, E, B, SA, O	N, SA	N, SA	N, SA, O	N, NAS, E, B, O	N, B, SA
GPx (U)	20.05 ± 2.5	36.91 ± 7.95	30.19 ± 7.38	38.9 ± 10.05	55.97 ± 7.89	32.1 ± 11.69
Signif. dif. (*p* ≤ 0.05)	NAS, E, B, SA, O	N, E, SA	N, NAS, B, SA	N, E, SA	N, NAS, E, B, O	N, SA
TAC (U/mL)	36.32 ± 3.12	20.85 ± 5.45	21.66 ± 6.24	21.93 ± 5.34	17.7 ± 0.63	23.05 ± 13.15
Signif. dif. (*p* ≤ 0.05)	NAS, E, B, SA, O	N	N	N, SA	N, B	N
LPO (µmol/L)	0.15 ± 0.02	1.77 ± 0.93	1.04 ± 0.32	1.01 ± 0.86	4.55 ± 0.36	1.66 ± 1.42
Signif. dif. (*p* ≤ 0.05)	NAS, E, B, SA, O	N, E, B, SA	N, NAS, SA	N, NAS, SA	N, NAS, E, B, O	N, SA
NO (µmol/L)	6.72 ± 2.02	30.25 ± 9.83	18.52 ± 6.59	32.41 ± 7.24	40.7 ± 4.37	20.8 ± 8.58
Signif. dif. (*p* ≤ 0.05)	NAS, E, B, SA, O	N, E, SA, O	N, NAS, B, SA	N, E, SA, O	N, NAS, E, B, O	N, NAS, B, SA
8-OHdG (ng/mL)	1.70 ± 0.34	2.82 ± 1.37	3.59 ± 1.4	2.41 ± 0.71	6.36 ± 1.14	3.10 ± 1.87
Signif. dif. (*p* ≤ 0.05)	NAS, E, SA, O	N, E, SA	N, NAS, SA	SA	N, NAS, E, B, O	N, SA

Samples categorised according to the identified bacterial genus or species: N—negative samples, NAS—non-aureus staphylococci group, E—enterococci group, B—*Bacillus* spp. group, SA—*S. aureus* group, O—other pathogens group; CAT—catalase, GPx—glutathione peroxidase, TAC—total antioxidant capacity, LPO—lipid peroxides, NO—nitric oxide, 8-OHdG—8-hydroxy-deoxyguanosine; Signif. dif.: significantly different from.

## Data Availability

Not applicable.

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
