# Peer review of "Milk Pathogens in Correlation with Inflammatory, Oxidative and Nitrosative Stress Markers in Goat Subclinical Mastitis"

_animals, 2022, doi:10.3390/ani12233245_

Round 1
Reviewer 1 Report
Comments
- In the abstract it is necessary to state what was the breed of goats. The authors only wrote Carpatian milk, that is too general a term
- The introduction, the purpose of the research should be described in detail.
- The material of the method please complete the information about the weight of the goats, if they were all the same age (which lactation). What was the fertility of the goats (how many children they fed). These factors affect milk yield as well as mastitis. Whether the milk was taken in summer and winter (please complete this information). Were there samples of morning or evening milking?. If the milk samples were taken in summer and winter, please complete the description of what the winter diet was like
- The summary should indicate the practical aspect of the research and the possibility of introducing it into practice
Author Response
Dear reviewer,
Thank you for reviewing our manuscript and sending your valuable comments and suggestions. Please find our answers in the enclosed document.

Reviewer 2 Report
The limitation os this study is the number of collected samples& collecting samples from only one farm.
Lines51-53 please add some statistical data, and be aware of world leading trend of avoiding proteins of animal origin in the diet - rising popularity of plant based drinks, and plant based alternative protein sources.
Line 71 raw milk/products from raw milk
About the differences between goat& cow milk the reviewer adivices add reference: http://doi.org/10.1111/1750-3841.15574; Prosser 2021
Some of the references Authors are using are quite old - please check &verify more actual literature data - it shows, that the problem rised in the manuscript is actual and important. e.g line 119-122, please check: Podhoercka et al. 2021 doi: 10.3390/foods10051046;
Gecaj et al. 2021 https://doi.org/10.3389/fvets.2021.69411
Kuchtik et al. 2021 10.1080/1828051X.2021.1913077
Lianu et al. 2021
line 174; only animals in 1st lactation were enroled in the study?
prevalence of mastitis in the herd? milk yield of animals? CMT or WST done?
The reviewer does not see the statistics on graphs and in the tables. Ststistical multiple comparison test?
Please add basic analysis of milk samples (% protein, % fat etc.)
TBC- was it checked?
The reviewer has doubts to Table 2 data; please check SD
What was the control in this study? or the stattistics was based of mullitiple comparisons?
There is no need to double the presentation of the results - please choose table or graphic presenation (Table 4 or Fig1)
What tests were done for analysis the milk sensory/organoleptic parameters?
Please elaborate more the part of correlations.
lines 529-532; please check the law regulations: In the United States of America, there is relevant legislation for goat milk , allowing up to 0.75 x 106 cells mL-1 in the bulk-tank milk.
The European Union legislation:has set a threshold for total bacterial counts in the bulk-tank milk of small ruminants equal to 1,500,000 colony-forming units (cfu) mL-1 for milk that would undergo thermal processing and 500,000 cfu mL-1 for milk that would be used for direct consumption (Reg. 853/2004).
And please remember you were working on individual milk samples and regulations are dedicated to bulk milk!
Please make conclusions more brief. Some part can be moved up to discussion part.
The Authors didnt discuss the aspect of milking and environment hygiene in contects of microbs. There is no information about regular processing of this milk.
Please highlight the novelty & strenghts of this study
Author Response
Dear reviewer,
Thank you for sending your valuable comments and suggestions and for taking the time to review our manuscript. Please find enclosed our responses/comments.

Reviewer 3 Report
Please see the attached file for comments and suggestions to the authors

Author Response
Dear reviewer,
Thank you for reviewing our manuscript and sending your valuable comments and suggestions that helped us improve it. Please find enclosed our point-by-point responses.

Round 2
Reviewer 2 Report
lines 31-32, please make it more clear "assessment of SCC, any of evaluated biochemical.."
line 60 please check the numbers in citations brackets
Author Response
Dear reviewer,
Thank you for your valuable comments, please find enclosed our responses to your suggestions.
